# Prevalence of Antimicrobial Resistance in *Escherichia coli* and *Salmonella* Species Isolates from Chickens in Live Bird Markets and Boot Swabs from Layer Farms in Timor-Leste

**DOI:** 10.3390/antibiotics13020120

**Published:** 2024-01-25

**Authors:** Abrao Pereira, Hanna E. Sidjabat, Steven Davis, Paulo Gabriel Vong da Silva, Amalia Alves, Cristibela Dos Santos, Joanita Bendita da Costa Jong, Felisiano da Conceição, Natalino de Jesus Felipe, Augusta Ximenes, Junilia Nunes, Isménia do Rosário Fária, Isabel Lopes, Tamsin S. Barnes, Joanna McKenzie, Tessa Oakley, Joshua R. Francis, Jennifer Yan, Shawn Ting

**Affiliations:** 1Global and Tropical Health Division, Menzies School of Health Research, Charles Darwin University, Dili, Timor-Leste; 2Ministry of Agriculture, Livestock, Fisheries and Forestry, Government of Timor-Leste, Av. Nicolao Lobato, Comoro, Dili, Timor-Leste; 3Epivet Pty. Ltd., Withcott, QLD 4352, Australia; 4School of Veterinary Science, Massey University, Palmerston North 4442, New Zealand

**Keywords:** antimicrobial resistance, chickens, Timor-Leste, *E. coli*, *Salmonella* species, local chicken, layer, broiler, boot swab, cloacal swab

## Abstract

The rapid emergence of antimicrobial resistance is a global concern, and high levels of resistance have been detected in chicken populations worldwide. The purpose of this study was to determine the prevalence of antimicrobial resistance in *Escherichia coli* and *Salmonella* spp. isolated from healthy chickens in Timor-Leste. Through a cross-sectional study, cloacal swabs and boot swabs were collected from 25 live bird markets and two layer farms respectively. *E. coli* and *Salmonella* spp. from these samples were tested for susceptibility to six antimicrobials using a disk diffusion test, and a subset was tested for susceptibility to 27 antimicrobials using broth-based microdilution. *E. coli* and *Salmonella* spp. isolates showed the highest resistance towards either tetracycline or ampicillin on the disk diffusion test. *E. coli* from layer farms (odds ratio:5.2; 95%CI 2.0–13.1) and broilers (odds ratio:18.1; 95%CI 5.3–61.2) were more likely to be multi-drug resistant than those from local chickens. Based on the broth-based microdilution test, resistance to antimicrobials in the Timor-Leste Antimicrobial Guidelines for humans were low, except for resistance to ciprofloxacin in *Salmonella* spp. (47.1%). Colistin resistance in *E. coli* was 6.6%. Although this study shows that antimicrobial resistance in chickens was generally low in Timor-Leste, there should be ongoing monitoring in commercial chickens as industry growth might be accompanied with increased antimicrobial use.

## 1. Introduction

Antimicrobial resistance (AMR) is a major global challenge, and its rapid emergence is attributed to the inappropriate and excessive use of antimicrobials in humans and animals [1]. Antimicrobials have been used in food-producing animals for improving weight gain, improving feed efficiency and managing diseases [1], with the projected usage expected to rise over the next decade [2]. The impacts of AMR include higher treatment costs, higher treatment failure and reduced animal production [3]. It is estimated that AMR will result in approximately 10 million human deaths per year and a global economic loss around USD 2 trillion per year by 2050 [4], with impacts being more pronounced in low- and middle-income countries due to their weaker public health systems [5].

Timor-Leste is a developing country located in Southeast Asia with a total population in 2022 of 1,340,434 [6]. Animal production is predominantly small-scale [7] and is an integral part of livelihood for most of the rural population [8,9]. Chickens are the most abundant food-producing animals in the country [8], and eggs are one of the most important sources of animal protein [10,11]. Chickens are also important for cultural purposes [12], and most households in Timor-Leste own chickens [8,10]. The chicken population in Timor-Leste can be divided into four types with different husbandry and management practices that might influence antimicrobial usage and hence AMR prevalence—local chickens, fighting cocks, broilers and layers [12]. Local chickens are mostly native chickens that are often raised in backyard settings by smallholder households for egg and meat production [10,13]. Fighting cocks are native roosters that are used for cock fighting. They can have a high market value, receive medical treatment for sustained injuries, and could ultimately be consumed as food [7,12]. Layer and broiler chickens are farmed commercially from imported day-old chicks [12,14], with production increasing in recent years in line with trends in other developing countries [12,15]. There are two large commercial layer farms in the country, but the exact number of broiler farms in the country is uncertain. All four chicken types can be found in live bird markets (LBMs) with varying availability. In LBMs close to the border with Indonesia, live broilers farmed in Indonesia can also be found [12].

All antibiotics in Timor-Leste are imported, and the most commonly imported classes of veterinary antibiotics are tetracyclines, penicillins, aminoglycosides and sulfonamides [14]. The use of antibiotics in animals in Timor-Leste is reportedly low compared to other countries due to the predominately subsistence production system [14]. The largest importer of veterinary antibiotics is the government [14], and these antibiotics are used by government animal health workers who provide free animal health services to smallholder farmers [16,17]. The second largest importers of veterinary antibiotics are commercial layer and broiler farms, where almost all imported antibiotics are meant for oral administration [14].

In the context of chickens, AMR in gastrointestinal bacteria such as *E. coli* and *Salmonella* spp. is relevant for both animal health and public health [18,19,20]. *E. coli* can cause gastrointestinal and extraintestinal infections such as urinary tract infections and blood-stream infections in humans [21]; and avian pathogenic *E. coli* can cause colibacillosis in chickens [19]. *Salmonella* spp. can cause gastrointestinal disease in chickens and humans, as well as significant production loss in chickens due to pullorum disease and fowl typhoid [19,22]. Several studies in other countries have shown a high prevalence of AMR in bacteria isolated from chickens, where resistance to tetracycline, penicillin and sulfonamide were often above 80% from bacteria isolated from commercial farms [23,24,25,26]. Even in free-range chickens from rural areas of Bangladesh without recorded antibiotic usage, extended-spectrum β-Lactamase (ESBL) producing *E. coli* was detected in almost a quarter of the tested faecal samples [27].

In Timor-Leste, there have been some studies on AMR in humans that have shown relatively high rates of carriage of resistant bacteria in humans exhibiting resistance to critically important antimicrobials such as ceftriaxone, ciprofloxacin and gentamicin [28,29]. However, there are no published studies investigating the role of animals in the transmission of antimicrobial resistant bacteria and its potential public health implications. This study focuses on chickens as a potential source of AMR, especially with the expansion of chicken production in the country, which could be accompanied with increase antimicrobial use [30]. This study has the primary objective of estimating the prevalence of AMR in *E. coli* isolated from cloacal swabs from healthy chickens in LBMs and boot swabs from layer farms in Timor-Leste, and a secondary objective of estimating the prevalence of AMR in *Salmonella* spp., isolated from the same samples. The study will also provide a baseline for monitoring AMR trends in the future.

## 2. Materials and Methods

### 2.1. Study Area and Population

A cross-sectional study was conducted in LBMs and layer farms in all 13 municipalities of Timor-Leste from August 2021 to February 2023.

### 2.2. Data Collection from Live Bird Markets

The target population for sampling was chickens sold in LBMs. The definition of LBMs is an outdoor space with two or more chicken vendors or traders where live chickens are sold at least once a week [31]. LBMs are typically located in the capital city of municipalities or town centres of sub-municipalities in Timor-Leste. These LBMs typically function as a collection point for chickens produced in nearby areas. Local chickens and fighting cocks are the main types of chickens found at LBMs, although broilers and spent hens can be found occasionally after the end of their production cycles. All chickens are sold live by vendors, and no slaughter occurs at these locations. Customers at these markets include households and food establishments.

In this study, a minimum sample size of 267 *E. coli* isolates was determined using the Epitools sample size calculator for estimating a proportion using the following parameters assuming there was no clustering: estimated antimicrobial resistance prevalence of 50%, 6% desired precision and 95% confidence level. The minimum sample size for cloacal swabs from chickens from LBMs was adjusted upwards to 334, based on an estimated 80% laboratory recovery rate for *E. coli*. The initial plan was to allocate samples to municipalities proportionate to the size of the human population. However, the availability of teams to visit municipalities was influenced by research and training activities, resulting in over/under representation of samples from some municipalities. Due to the lack of a central registry of LBMs in the country, a sampling frame was developed in collaboration with the Timor-Leste Ministry of Agriculture, Livestock, Fisheries and Forestry (MALFF) animal health workers working in each municipality. Details of LBM location, size, operating days and hours were collected. The sampling team visited every municipality for a period of up to 3 days, during which they visited the largest LBM that was operational and approximately within 1 hour driving time from the capital city of that municipality. If time permitted, the next largest LBMs that fulfilled the same criteria from each municipality were also visited. This process continued until the target sample size was achieved, with multiple visits to some municipalities.

Visits to most LBMs were with a local government animal health worker who facilitated the initial communication with chicken vendors. Within each LBM, all vendors selling chickens were approached, but only one healthy chicken from each type of chicken (local chicken, fighting cock, broiler and layer) was selected per chicken vendor to reduce the effect of clustering assuming that chickens from the same vendor could have originated from the same farm. The selection of chickens from each vendor was random. When a vendor’s chickens were in an inaccessible cage area, random sampling was only applied to the subgroup of chickens in accessible cage areas. A cloacal sample using an Amies swab (Copan, Italy) was taken from each selected bird, and each swab was labelled. Information on each chicken from which the cloacal sample originated was collected using the REDCap mobile app on Android tablets [32,33]. The sample number, name of chicken vendor, name of the LBM, type of chicken selected (i.e., local chicken, layer, broiler or fighting cock) were recorded.

### 2.3. Data Collection from Layer Farms

There are two large commercial layer farms in Timor-Leste [14]. Only layer houses that contained adult birds (above 20 weeks of age) within these two farms were targeted for sampling. A minimum sample size of 97 *E. coli* isolates was determined using the Epitools sample size calculator for estimating a proportion using the following parameters assuming there was no clustering: estimated antimicrobial resistance prevalence of 50%, 10% desired precision and 95% confidence level. Sampling was conducted proportionate to size, according to the total chicken population on each layer farm. This resulted in a minimum of 59 samples and 38 samples collected from the larger and the smaller farms, respectively.

Prior to sampling, the number of houses with adult layer birds was determined with the farm manager from each farm. Up to four environmental boot swabs were collected from each house, with the number of boot swabs collected from a house reflecting the size of the chicken population. Commercial boot swabs premoistened with skim milk (Romer Labs, Getzersdorf, Austria) were used for sample collection. Sample collection was performed by walking along the outer faecal lanes underneath each layer house, as the middle faecal lanes were often difficult to access. For houses with cages that were not raised sufficiently high enough for an operator to walk underneath, the boot swab was attached securely to a stick to collect faecal material from at least 10 different locations along the entire length of the faecal lane. Each pair of boot swabs was placed into a sterile plastic bag and labelled. Information on each house from which the boot swab originated from was collected using the REDCap mobile app on Android tablets [32,33]. The house number, source of birds, age of birds, vaccination status and history of antimicrobial use were collected.

### 2.4. Isolation and Identification of Bacteria

All samples were stored and transported in a cool box with ice packs to the Veterinary Diagnostic Laboratory (VDL) in Dili, the capital of Timor-Leste. The VDL is the only animal health laboratory in the country and is a government laboratory under the MALFF. When samples were not delivered within 24 h of collection, they were stored in refrigerated conditions at the local government office or accommodation. All cloacal swabs and boot swab samples were processed within 7 days upon arrival at VDL and screened for *E. coli* and *Salmonella* spp. Antimicrobial susceptibility testing (AST) was performed on the isolates followed by storage in a −80 °C biorepository.

For isolation of *E. coli* and *Salmonella* spp., Amies cloacal swabs and boot swabs were pre-enriched in 10 mL and 200 mL of buffered peptone water (BPW), respectively, and incubated at 37 °C for 18–24 h. After incubation, a loopful (10 µL) of overnight BPW culture was streaked onto selective Brilliance *E. coli* agar (Oxoid, Thermo Fisher, Waltham, MA, USA) and incubated at 37 °C for 18–24 h to isolate *E. coli*. At the same time, 1 mL of overnight BPW culture was added to 10 mL of Rappaport–Vassiliadis soy broth (RVS broth) (Oxoid, Thermo Fisher) for 18–24 h at 41 °C for enrichment of *Salmonella* spp. After enrichment, 10 µL of RVS broth was streaked onto Xylose Lysin Deoxycholate agar and incubated at 37 °C for 24 h (Becton Dickinson, East Rutherford, NJ, USA). One suspected *E. coli* and *Salmonella* spp. isolate from each sample were propagated on different nutrient agar plates for biochemical identification using Microbact 12A (Oxoid, Thermo Fisher). All isolates identified as *Salmonella* spp. on Microbact 12A were confirmed using a salmonella latex agglutination test following the manufacturer’s instructions (Oxoid, Thermo Fisher).

### 2.5. Antimicrobial Susceptibility Testing

Antimicrobial susceptibility testing was performed using the disk diffusion test (DDT) on all isolated *E. coli* and *Salmonella* spp. according to the Clinical and Laboratory Standard Institute (CLSI) testing methodology for DDT [34].

For each isolate, a standardised inoculum with turbidity of 0.5 McFarland was streaked onto Mueller–Hinton agar (Becton Dickinson, USA). The six antimicrobial disks used were ampicillin (AMP; 10 µg), streptomycin (STR; 10 µg), tetracycline (TET; 30 µg), trimethoprim/sulfamethoxazole (SXT; 1.25/23.75 µg), sulfisoxazole (SOX; 250 µg) and enrofloxacin (ENR; 5 µg). These six antimicrobial disks were selected based on commonly used antimicrobials in animals in Timor-Leste [14] and cover five classes of antimicrobials—namely penicillins, aminoglycosides, tetracyclines, fluoroquinolones and sulfonamides. Trimethoprim/sulfamethoxazole and sulfixosazole were considered to be within the same sulfonamide antimicrobial class. The DDT petri dishes were incubated for 18 to 24 h at 37 °C. The zone of inhibition in millimetres was measured under a magnification glass and recorded on the WHONET 5.6 software. The observed inhibition zones were interpreted according to CLSI interpretative criteria for Enterobacterales [34]. Internal quality control for the DDT was performed using *E. coli* ATCC 25922. External quality assurance was conducted through a program called EQAsia, which is supported by the Fleming Fund. 

To understand the resistance profile of isolates to antimicrobials relevant for human health, a subset of isolates was transferred from the VDL to the National Health Laboratory (NHL) in Timor-Leste for AST using a broth-based microdilution method. The NHL is a government human health laboratory that provides reference and clinical microbiological services [35]. These isolates were tested with a panel of 27 antimicrobials on the NMIC-502 (Becton Dickinson, East Rutherford, NJ, USA) using the BD Phoenix M50 instrument at the NHL. This also consisted of the Phoenix ESBL test (BD Diagnostic Systems, Sparks, Md) and Phoenix CPO detect test (BD Diagnostic Systems, Sparks, Md). The Phoenix ESBL test determines whether an isolated is an ESBL based on phenotypic response to a panel of cephalosporins, which are alone or in combination with clavulanic acid, while the Phoenix CPO detect test identifies carbapenemase-producing bacteria of classes A, B and D. The raw minimum inhibitory concentration (MIC) data were entered into the WHONET software. The results were interpreted according to European Committee on Antimicrobial Susceptibility Testing (EUCAST) clinical breakpoints version 13.0 [36], except for nitrofurantoin breakpoints in *Salmonella* spp., which were interpreted according to CLSI M100 interpretative criteria for Enterobacterales [37] because none were available in EUCAST.

### 2.6. Data Analysis

All results were validated and checked for missing entries and entry errors. The bacterial recovery rate was determined by the number of samples from which each target bacteria was isolated divided by the total number of tested samples. Descriptive statistics were used to calculate the proportion of resistance to different antimicrobials amongst the *E. coli* and *Salmonella* spp. isolates tested with the DDT and broth-based microdilution method. Data were presented based on the origin of the isolates, namely LBM or layer farm. For the DDT results, the percentage of resistance was estimated with 95% confidence intervals (CI) adjusted for clustering by municipality and house, respectively. Multi-drug resistance (MDR) was determined based on the DDT and broth-based microdilution results. An isolate was classified as MDR if it was resistant to three or more antimicrobial classes. The phenotypic resistance profiles of *E. coli* and *Salmonella* spp. isolates were described.

The calculation of MIC50 and MIC90, which is the MIC value able to inhibit ≥ 50% and ≥90% of isolates, respectively [38], was performed on the *E. coli* isolates that underwent broth-based microdilution. Isolates that were resistant to ceftriaxone (MIC > 1 mg/L) and ceftazidime (MIC > 1 mg/L) were considered as potential ESBL producers [39]. Isolates that had meropenem MIC > 0.125 mg/L were considered as potential carbapenemase producers [39].

Mixed-effect logistic regression models were used to assess the crude associations between the source of *E. coli* isolates (i.e., local chicken, fighting cock, broiler chicken and layer farm) and resistance to each antimicrobial using both the DDT and broth-based microdilution results. The outcome variable was resistant status versus susceptible and intermediate. The source of the isolate (i.e., municipality or layer farm) was fitted as the random effect to account for clustering. Data analyses were performed using Stata 17.0 [40].

## 3. Results

### 3.1. Origin of Samples and Recovery of Isolates

In total, 345 cloacal swab samples were collected from 25 LBMs located in 13 municipalities, comprising samples from 254 local chickens, 72 fighting cocks, 17 broiler chickens and 2 layer chickens. During the sampling period, broiler and layer chickens were only found in LBMs located in Bobonaro or Dili municipalities. For LBM sampling, allocation to municipalities was generally proportionate to size. However, Bobonaro municipality was over-represented, and Dili municipality was under-represented. The number of cloacal swab samples collected from each municipality is shown in Figure 1. A detailed breakdown of the number of LBMs, number of cloacal swabs collected, and type of chicken from which cloacal swabs samples originated can be found in Appendix A. A total of 87 boot swab samples were collected from two large commercial layer farms, with 57 samples from the larger farm and 30 samples from the smaller farm. At the time of sampling, the estimated total number of birds reported by the larger and smaller farms were 113,836 and 80,000, respectively. Both layer farms vaccinated chickens against Newcastle disease. Only one farm reported using antibiotics, specifically an oxytetracycline injectable antibiotic used only in sick birds.

For LBMs, the recovery rates for *E. coli* and *Salmonella* spp. from 345 chicken cloacal swabs were 85.5% (295 isolates) and 2.3% (8 isolates), respectively. For layer farms, the recovery rate of *E. coli* and *Salmonella* spp. from 87 boot swabs were 85.1% (74 isolates) and 33.3% (29 isolates), respectively. The number of isolates of *E. coli* and *Salmonella* spp. from LBMs by type of chicken and layer farms along with the recovery rate can be found in Table 1.

### 3.2. Disk Diffusion Results

The prevalence of antimicrobial resistance in *E. coli* and *Salmonella* spp. isolates with 95% confidence intervals based on the location from which the samples were collected (LBMs and layer farms) can be found in Table 2. The *E. coli* isolates from chickens in LBMs showed the highest resistance levels towards ampicillin and tetracycline. The *E. coli* isolates from layer farms showed the highest resistance levels towards tetracycline and sulfisoxazole. The *E. coli* isolates from both LBMs and layer farms had the lowest resistance to enrofloxacin compared to other antimicrobials. Multi-drug resistance was present in 7.8% (95%CI: 5.3–11.4) of *E. coli* isolates from LBMs and 20.3% (95%CI: 9.7–37.5) of *E. coli* isolates from layer farms. The prevalence of AMR in *E. coli* based on origin is shown in Figure 2 and Appendix A. For each tested antimicrobial, the prevalence of resistance in *E. coli* isolates generally increased across local chickens, fighting cocks, layer farms and broilers. The *Salmonella* spp. isolates originating from both LBMs and layer farms showed the highest resistance to tetracycline. A total of 7 out of 37 *Salmonella* spp. isolates (18.9%) were identified as MDR.

The phenotypic resistance profile of all *E. coli* and *Salmonella* spp. isolates are shown in Table 3. The proportion of *E. coli* and *Salmonella* spp. Isolates that were not resistant to any of the tested antimicrobials on the DDT were 72.2% (213/295 isolates) and 25.0% (2/8 isolates), respectively. None of the *E. coli* isolates were resistant to all six antimicrobials, 3.3% (12/369 isolates) were resistant to five antimicrobials, 3.8% (14/369 isolates) to four antimicrobials, and 6.8% (25/369 isolates) to three antimicrobials. The most common phenotypes with resistance to at least one antimicrobial was TET (17/369 isolates) and AMP-TET (17/369 isolates). The predominant phenotypic resistance profiles of *E. coli* isolates resistant to three or more antimicrobials were TET-SXT-SOX (10/369 isolates), AMP-STR-TET-SXT-SOX (8/369 isolates) and AMP-TET-SXT-SOX (6/369 isolates). For *Salmonella* spp. isolates, 5.4% (2/37 isolates) were resistant to all tested antimicrobials, 5.4% (2/37 isolates) were resistant to five antimicrobials, and 8.1% (3/37 isolates) were resistant to four antimicrobials. The predominant phenotypic resistance profile of *Salmonella* spp. isolates resistant to three or more antimicrobials was AMP-STR-TET-SOX (3/37 isolates).

### 3.3. Broth-Based Microdilution Results

A subset of 212 *E. coli* and 17 *Salmonella* spp. isolates were tested for resistance to an extended panel of antimicrobials using broth-based microdilution. The percentage of resistance, MIC50 and MIC90 to various antimicrobials for the 212 *E. coli* isolates sent for broth-based microdilution is presented in Table 4. The results for tigecycline and ceftazidime-avibactam using the BD Phoenix M50 instrument has not been validated for use by the NHL and was excluded from analysis. The *E. coli* isolates tested using broth-based microdilution showed elevated resistance levels to ampicillin (22.2%), amoxicillin-clavulanic acid (20.3%), piperacillin (17.0%) and temocillin (10.4%), which belongs to the penicillin or β-lactam/β-lactam inhibitor combination class of antimicrobials. Resistance to all other antimicrobials was below 10%. Resistance to colistin was 6.6%. Except for ampicillin, amoxicillin-clavulanic acid and trimethoprim-sulfamethoxazole, all antimicrobials included in the Timor-Leste Antimicrobial Guidelines for humans [41] have resistance levels below 6%. There was no resistance to amikacin, meropenem and ertapenem. Eight *E. coli* isolates (3.8%) were determined to be potential ESBL-producers based on EUCAST guidelines [39], but none were considered as ESBL-positive isolates based on the Phoenix ESBL test. A total of 21 *E. coli* isolates (9.9%) were potential carbapenemase producers based on EUCAST guidelines [39], but none were considered carbapenemase-producing organisms based on the Phoenix CPO test. Multi-drug resistance was present in 13.7% (29/212) of *E. coli* isolates that underwent broth-based microdilution.

The percentages of resistance to various antimicrobials for the 17 *Salmonella* spp. isolates sent for broth-based microdilution is presented in Table 5. All but 1 of the 17 isolates originated from layer farms. The highest resistance levels were observed in fluoroquinolones, such as ciprofloxacin (47.1%), and penicillins, such as temocillin (41.2%). No or very low resistance was observed to all other classes of antimicrobials. Multi-drug resistance was present in 5.9% (1/17) of the *Salmonella* spp. isolates that underwent broth-based microdilution, and this isolate originated from a local chicken in a live bird market located in a municipality along the border with Indonesia.

The proportion of *E. coli* and *Salmonella* spp. isolates that were not resistant to any antimicrobials based on broth-based microdilution were 59.4% (126/212 isolates) and 29.4% (5/17 isolates), respectively. The phenotypic resistance profiles to different antimicrobial classes for multi-drug resistant *E. coli* and *Salmonella* spp. based on broth-based microdilution can be found in Appendix A. The majority of MDR isolates were resistant to both penicillins and β-lactam/β-lactam inhibitor combination of antimicrobial classes. The predominant phenotypic resistance profiles of *E. coli* isolates resistant to three or more antimicrobials were ampicillin-piperacillin-ampicillin/clavulanic acid (8/212 isolates) and ampicillin-piperacillin-mecillinam-ampicillin/clavulanic acid (4/212 isolates). The most common phenotypic resistance profile of *Salmonella* spp. isolates resistant to three or more antimicrobials was ampicillin-piperacillin-temocillin-ciprofloxacin (2/17 isolates).

### 3.4. Comparison of Antimicrobial Resistance in Different Chicken Populations

The logistic regression results to determine the crude associations between the origin of *E. coli* isolates and resistance to each antimicrobial on the DDT are presented in Table 6. The *E. coli* isolates originating from layer farms (odds ratio [OR]: 5.2; 95%CI 2.0–13.1) and broilers (OR: 18.1; 95%CI 5.3–61.2) were more likely than those from local chickens to be MDR. *E. coli* isolates originating from layer farms were more likely than those from local chickens to have resistance to all tested antimicrobials on the DDT except for ampicillin. The highest odds ratios were observed for enrofloxacin (OR: 11.2; 95%CI: 2.3–55.4) and tetracycline (OR: 6.4; 95%CI 3.5–11.7). *E. coli* isolates from broilers were more likely than those from local chickens to have resistance against all tested antimicrobials on the DDT except for enrofloxacin. The highest odds ratios were observed for sulfisoxazole (OR: 17.4; 95%CI 5.7–53.0), trimethoprim/sulfomethoxazole (OR: 8.7; 95%CI 2.8–27.4) and streptomycin (OR: 8.2, 95%CI 2.2–30.1). Point estimates suggested that *E. coli* isolates from fighting cocks were more likely than those from local chickens to have resistance against all tested antimicrobials except enrofloxacin, but estimates were imprecise so no definitive conclusion could be reached.

The logistic regression for determining the crude associations between the origin of *E. coli* isolates and resistance to each antimicrobial based on broth-based microdilution are presented in Appendix A. Broilers were not included in the analysis because there were only six isolates, of which all were susceptible to all antimicrobials tested except for one isolate that was resistant to gentamicin. The *E. coli* isolates from layer farms were much more likely to be resistant to ciprofloxacin (OR: 12.7; 95%CI: 1.4–117.2), cephalexin (OR: 5.4; 95%CI: 1.2–23.4) and trimethoprim-sulfamethoxazole (OR: 6.4; 95%CI: 2.0–20.2) compared with those from local chickens in LBMs. Although point estimates of the association for ampicillin, piperacillin, cefuroxime and levofloxacin were far from the null (OR > 2), the estimates were imprecise so no definitive conclusion can be reached. No definite conclusion could be reached on whether *E. coli* isolates from fighting cocks were more likely than those from local chickens to have resistance to all tested antimicrobials based on broth-based microdilution, although point estimates of the association for cefepime, cefixime, ceftazidime, ceftriaxone, cefuroxime, cephalexin, colistin and tobramycin were far from the null (OR > 2).

## 4. Discussion

### 4.1. Strength of the Study and Key Findings

This is the first study describing antimicrobial resistance in *Enterobacteriaceae* originating from chickens in Timor-Leste, and one of the few studies conducted in the context of a small developing country with limited resources. *E. coli* and *Salmonella* spp. isolates from chickens showed higher resistance levels to tetracycline and penicillin classes of antimicrobials compared to most other classes. These two classes of antimicrobials were imported in the highest quantities into Timor-Leste for use in animals based on import data between 2016 and 2019 [14] and are the two most commonly used classes of antimicrobials in chickens reported by government animal health workers between 2021 and 2022 [16]. These antimicrobials appear to be the main drivers of MDR bacteria in Timor-Leste.

### 4.2. Recovery Rate of Isolates

Commensal bacteria like *E. coli* can be found in the intestinal tract of all chickens [42]. Therefore, the Food and Agriculture Organization Regional Antimicrobial Resistance Monitoring and Surveillance Guidelines states that a 100% recovery rate for *E. coli* could be expected [18]. In this study, a relatively high *E. coli* recovery rate from cloacal swabs was achieved, likely due to the use of refrigerated sample transport and good bacterial isolation methods. This *E. coli* recovery rate was similar to the recovery rate from another study collecting cloacal swabs from broilers in China (81.6%) [43] and was much higher than other studies in Qatar (52.3%) [44] and Malaysia (51.8%) [25] where cloacal swabs were also collected.

In this study, there was a difference in the recovery rates for *Salmonella* spp. between samples originating from LBMs (2.3%) compared to layer farms (33.3%). The recovery rate for *Salmonella* spp. from cloacal swabs originating from LBMs is similar to those of other studies targeting village or backyard chickens in Malaysia (2.5%) [45], Paraguay (3.5%) [46] and Iran (5.8%) [47]. The recovery rate for *Salmonella* spp. from boot swabs from layer farms was similar to those of other studies [48,49,50]. The higher recovery rate for *Salmonella* spp. from layer farms compared to those from fighting cocks and local chickens from LBMs could be due to a higher prevalence of *Salmonella* spp. In commercial farms compared to backyard farming, where a higher population density can increase *Salmonella* spp. Transmission [51]. It could also be due to the use of boot swabs for sample collection on layer farms, which has been shown to yield a higher recovery rate for *Salmonella* spp. Compared to cloacal swabs [52,53], especially since shedding from the cloaca can be intermittent [54].

### 4.3. Antimicrobial Resistance in Different Chicken Populations

Antimicrobial resistance based on the DDT results for *E. coli* isolates from local chickens was lower than reported in other countries focusing on a similar target population of backyard chickens [55,56,57]. This could be due to the lack of access to government veterinary services in Timor-Leste [58], which is a common source of antimicrobials for most backyard farmers [17]. Similarly, antimicrobial resistance based on the DDT results for *E. coli* and *Salmonella* spp. isolates from layers and broilers was also lower than those reported in other Southeast Asian countries [25,59]. The lower antimicrobial resistance levels in commercial chickens in Timor-Leste compared to those of other countries could be due to the low availability of veterinary antimicrobials in the country, as there is no local manufacture, and all antimicrobials have to be imported [14]. The comparison of antimicrobial resistance results based on the DDT for *Salmonella* spp. to other studies should be undertaken cautiously given the small sample size and thus very wide 95% confidence intervals for resistance estimates. 

In this study, the percentage of *E. coli* isolates with MDR based on the DDT results was significantly higher in broiler chickens and layer farms than in local chickens. Specifically, the *E. coli* isolates originating from broiler chickens and layer farms were more likely to have higher resistance to several antimicrobials, such as tetracyclines, penicillins, fluoroquinolones and sulfonamides, compared to local chickens. For broilers, this may be due to the routine use of these antimicrobials during commercial farming, which has been reported to be common in neighbouring countries such as Indonesia [60,61]. The import of oral tetracyclines, penicillins and fluoroquinolones by broiler farms in Timor-Leste between 2016 and 2019 [14] provides some evidence of this practice, although actual data on antimicrobial usage by broiler farms was not available in this study because sampling of broilers occurred at LBMs. Some broilers collected from LBMs may have also originated from Indonesia through informal trade across the border [12] and reflect the higher resistance profiles found in the country of origin. One study on broiler chickens in Indonesia showed that *E. coli* resistance to tetracycline, ampicillin and ciprofloxacin were all above 80% [62]. To understand the antimicrobial resistance profiles of only broilers produced in Timor-Leste, future studies could consider sample collection directly from broiler farms, which would also allow the history of antimicrobial usage in broilers to be collected. 

It has been well documented that day-old chicks can act as a source of antimicrobial resistant bacteria for chicken farms [63,64,65,66]. Hence, another possible source of resistance for broilers and layers could be from day-old chicks, which are imported from Indonesia or Malaysia where antimicrobial resistance in commercial chickens is known to be high [25,59,62,67]. This may explain the resistance observed in the isolates from layer farms in this study, especially since there was no reported antimicrobial use in these farms except for injectable oxytetracycline in sick chickens. Therefore, the antimicrobial resistance profile of isolates from imported day-old chicks could be investigated through the sampling of day-old chicks at their point of entry into Timor-Leste.

### 4.4. Public Health Implications

Resistance to gentamicin and ciprofloxacin in *E. coli* isolates based on broth-based microdilution from this study was lower than that reported in *E. coli* isolates from the stools of healthy school children from Timor-Leste [29], although a direct comparison cannot be made because the screening of *E. coli* isolates in the latter study involved the use of selective ESBL agar [29]. No *E. coli* or *Salmonella* spp. isolates tested as ESBL positive or carbapenemase-producing organisms in this study. *E. coli* and *Salmonella* spp. resistance to antimicrobials listed in the Timor-Leste Antimicrobial Guidelines for human health [41] were generally below 25%, except for resistance to temocillin and ciprofloxacin in *Salmonella* spp. It is unlikely that the high rates of AMR carriage in humans in Timor-Leste [28,29] is driven by transmission from household chickens.

For colistin, a warning on the accuracy of several commercial systems for MIC results was previously issued by EUCAST, although it did not specifically evaluate the BD Phoenix M50 [68,69]. Recent studies indicate that the BD Phoenix M50 can produce accurate MIC results for colistin [70,71,72], and this has led to its inclusion in this study. The resistance to colistin in *E. coli* from chickens (6.6%) in this study was higher than that reported in many European countries where resistance is usually less than 1% [73,74,75], but resistance was lower than that reported in other parts of Asia [76]. Although colistin is not currently used in humans in Timor-Leste, resistance is still a concern because it is a drug of last resort against MDR Gram-negative bacteria [77]. It is also classified as a highest priority critically important antimicrobial by the World Health Organization [78], and a highly important antimicrobial agent by the World Organisation for Animal Health [79]. Oral colistin for use in chickens in Timor-Leste contributed to 4.8% of total antimicrobial imports between 2016 and 2019 [14] and could have contributed to the observed resistance. Antibiotic use guidelines for the commercial poultry industry can be developed to reduce AMR emergence to highest priority critically important antimicrobials for humans, such as colistin. For example, the usage of colistin could be limited for definitive treatment only when no other alternatives are available [80].

Ciprofloxacin is classified as a highest priority critically important antimicrobial by the World Health Organization [78] and is among the more commonly used antimicrobials in the health system in Timor-Leste [81]. Although the number of isolates tested was very small, a ciprofloxacin resistance of 50% among the 16 *Salmonella* spp. isolates originating from layer farms tested using broth-based microdilution was concerning. High ciprofloxacin resistance in *Salmonella* spp. from chickens has been reported to be an issue in other countries such as China [82,83], Bangladesh [84] and India [85] as well. Ciprofloxacin-resistant serovars, such as the *Salmonella enterica* serotype Kentucky from poultry is also a global concern [86,87,88], as they can also exhibit resistance to carbapenems and extended-spectrum cephalosporins [89]. However, none of the ciprofloxacin-resistant *Salmonella* spp. isolates from this study showed resistance to carbapenems and cephalosporins or were considered to be MDR. While there have been no imports of ciprofloxacin for use in animals into Timor-Leste, there have been imports of enrofloxacin, a veterinary fluroquinolone antimicrobial intended for use in commercial chicken production, between 2016 and 2019 [14]. This may have contributed to the elevated resistance to ciprofloxacin, as the use of enrofloxacin has been associated with increasing resistance to ciprofloxacin [24,90]. As the country continues to expand commercial chicken farming [14], the use of antimicrobials may potentially increase particularly if alternative disease control options are lacking for some poultry bacterial pathogens [30]. Therefore, ongoing antimicrobial resistance monitoring in the commercial chicken farming sector is recommended to detect emerging resistance patterns. 

### 4.5. Origin of Samples

The sample collection at LBMs was motivated by the recognition that these locations throughout Timor-Leste play an important role in the supply chain of chickens and in the potential dissemination of bacteria carrying resistance genes to consumers. Sampling from LBMs also means that swabs originate mostly from adult chickens after potential exposure to antimicrobials used during their production. Due to the convergence of chickens from multiple sources to a single point, sampling from LBMs is also known to be more practical and cost-effective especially in low-resource settings [91]. However, the downside of sample collection at LBMs is that it is often not possible to determine the farm-of-origin of the chickens and interact with farmers to understand antimicrobial use in their chickens. Taking these trade-offs into consideration, it is proposed that periodic LBM sampling could be used as the primary method for monitoring AMR trends in chickens in Timor-Leste due to its ability to collect useful data at a low cost, but this could be supplemented with additional sampling from commercial chicken farms and day-old chicks to investigate specific questions of interest.

### 4.6. Capacity Building and One Health

The animal health sector in Timor-Leste began investigating AMR in animals in 2019 as part of a capacity building programme that included side-by-side mentoring of local Timorese staff in study design, sample collection, sample transport, laboratory testing and data analysis. Rapid capacity development was achieved especially in the VDL through the permanent placement of experienced laboratory scientists to work alongside local Timorese staff over the past four years. Staff from the VDL also benefited from work placements in other government laboratories that routinely performed antimicrobial susceptibility testing in Australia (Berrimah Veterinary Laboratory) and Timor-Leste (NHL). Initial challenges related to limited access to consumables at the VDL was addressed through resource sharing with the NHL. A One Health approach was also utilised in strengthening antimicrobial resistance monitoring [92], especially in the areas of AMR data collection, analysis and communication for human and animal health. This includes the standardisation of AMR testing methods by using the same broth-based microdilution test to improve the comparability of information produced and promoting data sharing and integration through regular scientific meetings. 

### 4.7. Future Work

In this study, AMR monitoring in chickens focused on *E. coli* as an indicator bacteria and *Salmonella* spp. as a foodborne zoonotic pathogen. Further characterisation of *Salmonella* spp., especially those found in commercial poultry production could be considered. This would help determine the diversity of serotypes with zoonotic significance, such as *S. enterica* serovar Typhimurium and *S. enterica* serovar Enteriditis [93], and identify serotypes that can cause severe production loss in chickens such as *S. pullorum* [94]. This would also provide a deeper understanding on the epidemiology of salmonellosis in commercial chicken farms in Timor-Leste, including possible transmission pathways [95]. In addition, AMR monitoring could expand to more animal species, such as cattle, pigs and buffalo, as government animal health workers in Timor-Leste have reported administering antibiotics to these species most frequently [16].

Through further laboratory capacity building, AMR monitoring in animals has been expanding to include other bacteria of public health significance such as *Enterococcus* spp. and *Campylobacter* spp. Matrix-assisted laser desorption ionization time-of-flight mass spectrometry (MALDI-TOF MS), which is available at the NHL, could also be utilised for rapid bacterial identification and the further characterisation of isolates [96,97]. The panel of antimicrobials used in this study could be refined to closely harmonise with international guidelines for AMR testing in healthy animals [18], supplemented with the molecular characterization of isolates to identify potential emerging AMR threats.

## 5. Conclusions

This is the first study investigating AMR in animals in Timor-Leste that focused on *E. coli* and *Salmonella* spp. originating from chickens. Among the panel of antimicrobials tested, the highest levels of resistance were observed in antimicrobials belonging to the tetracycline and penicillin classes of antimicrobials, which is consistent with them being the most imported veterinary antimicrobials in the country. The isolates from local chickens had low levels of resistance, and this level of resistance was even lower than what was observed in backyard chickens from other countries. The isolates from commercial layer farms and broilers exhibited a higher level of resistance to several antimicrobials compared to local chickens. The source of resistance in commercial chickens could be attributed to on-farm antimicrobial use or imported day-old chicks. This should be further investigated through sampling from chickens directly at broiler farms, which would allow the concurrent collection of information on antimicrobial usage and sampling from imported day-old chicks to determine the carriage of antimicrobial resistant bacteria. Due to the expected continual growth of commercial chicken production and accompanying potential for increased antimicrobial use, there should be ongoing AMR and antimicrobial use monitoring in commercial chickens.

## Figures and Tables

**Figure 1 antibiotics-13-00120-f001:**
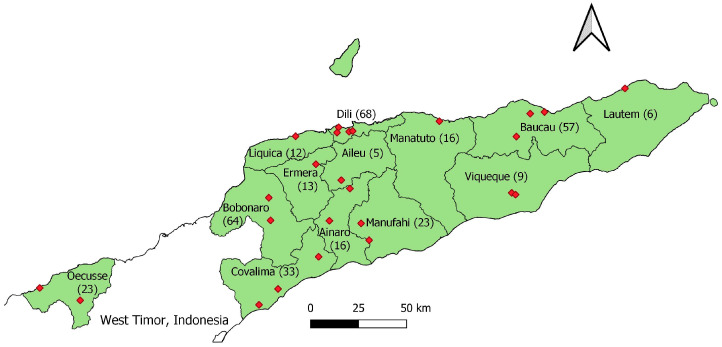
Map of Timor-Leste showing the 13 municipalities from which cloacal swab samples were collected. Numbers in brackets after each municipality name indicate the number of cloacal swab samples collected in each location. The red diamonds represent individual live bird markets where samples were collected.

**Figure 2 antibiotics-13-00120-f002:**
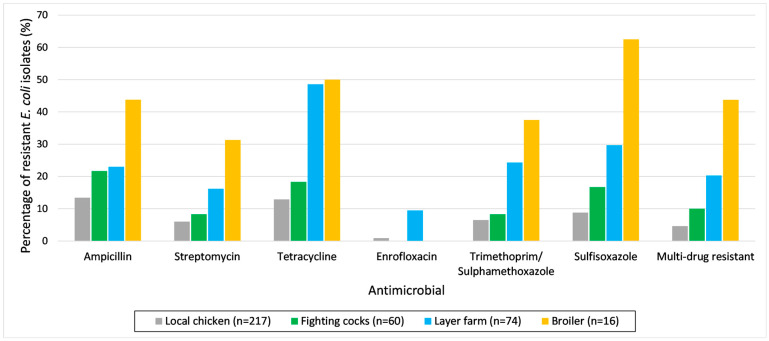
Percentage of *E. coli* isolates resistant to different antimicrobials from disk diffusion and multi-drug resistance based on the origin from which they were collected. Local chicken, fighting cock and broiler chicken samples were collected from live bird markets (LBMs). The percentage of resistance of *E. coli* isolates originating from layer chickens from LBMs is not shown because there were only two isolates.

**Table 1 antibiotics-13-00120-t001:** Number of isolates and recovery rates of *E. coli* and *Salmonella* spp. collected from different chicken types in live bird markets and layer farms in Timor-Leste from August 2021 to February 2023.

Location of Sampling	Origin of Sample	Number of Samples	*E. coli*		*Salmonella* spp.	
			Number of Isolates	Recovery Rate (%)	Number of Isolates	Recovery Rate (%)
Live bird markets	Local chicken	254	217	85.4	4	1.5
	Fighting cock	72	60	83.3	1	1.3
	Broiler	17	16	94.1	3	17.6
	Layer	2	2	100	0	-
	Total	345	295	85.5	8	2.3
Layer farms	Layer house	87	74	85.1	29	33.3

**Table 2 antibiotics-13-00120-t002:** Prevalence of AMR for *E. coli* and *Salmonella* spp. isolates from disk diffusion results based on location of collection in live bird markets (LBMs) or layer farms (LFs).

Antimicrobial	*E. coli* Isolates		*Salmonella* spp. Isolates	
	Prevalence from LBMs (*n* = 295) (%) (95%CI)	Prevalence from LF (*n* = 74) (%) (95%CI)	Prevalence from LBMs (*n* = 8) (%) (95%CI)	Prevalence from LF (*n* = 29) (%) (95%CI)
Ampicillin	16.6 (13.0–21.0)	23.0 (13.7–35.8)	25.0 (1.7–86.2)	24.1 (8.5–52.2)
Streptomycin	7.8 (4.4–13.3)	16.2 (8.9–27.8)	25.0 (1.7–86.2)	17.2 (3.4–55.0)
Tetracycline	16.3 (12.2–21.3)	48.6 (38.4–60.0)	75.0 (19.2–97.4)	31.0 (12.3–59.2)
Enrofloxacin	0.7 (0.2–2.0)	9.5 (4.2–20.0)	25.0 (1.7–86.2)	20.7 (9.5–39.3)
Trimethoprim/Sulphamethoxazole	8.8 (5.5–13.8)	24.3 (15.6–35.9)	25.0 (1.7–86.2)	3.4 (0.8–14.0)
Sulfisoxazole	13.6 (8.1–21.7)	29.7 (19.6–42.4)	37.5 (5.5–86.2)	20.7 (6.3–50.2)
Multi-drug resistant	7.8 (5.3–11.4)	20.3 (9.7–37.5)	25.0 (1.7–86.2)	17.2 (3.4–55.0)

**Table 3 antibiotics-13-00120-t003:** Phenotypic resistance profiles of *E. coli* (*n* = 369) and Salmonella (*n* = 37) isolated from live bird markets (LBMs) and layer farms (LFs) in Timor-Leste, to tested antimicrobials using disk diffusion.

Number of Antimicrobials	Phenotypic Resistance Profile	*E. coli*			*Salmonella* spp.		
		Percent of Isolates from LBMs (*n* = 295)	Percent of Isolates from LFs (*n* = 74)	Percent of Isolates from LBMs and LFs (*n* = 369)	Percent of Isolates from LBMs (*n* = 8)	Percent of Isolates from LFs (*n* = 29)	Percent of Isolates from LBMs and LFs (*n* = 37)
0	None	72.2	39.2	65.6	25.0	48.3	55.2
1	AMP	3.1	2.7	3.0	0	0	0
	STR	1.4	1.4	1.4	0	0	0
	TET	2.0	14.9	4.6	37.5	13.8	24.1
	SXT	0.3	0	0.3	0	0	0
	ENR	0.3	1.4	0.5	0	10.3	10.3
	SOX	2.4	0	1.9	0	3.4	3.4
2	AMP-TET	5.1	2.7	4.6	0	0	0
	AMP-SXT	0.3	0	0.3	0	0	0
	AMP-ENR	0	0	0	0	6.9	6.9
	STR-TET	0	2.7	0.5	0	0	0
	STR-ENR	0	1.4	0.3	0	0	0
	STR-SOX	1.4	0	1.1	0	0	0
	TET-ENR	0	2.7	0.5	0	0	0
	TET-SOX	0.7	0	0.5	12.5	0	3.4
	SXT-SOX	0.7	2.7	1.1	0	0	0
3	AMP-STR-TET	0.3	1.4	0.5	0	0	0
	AMP-STR-SOX	0.3	0	0.3	0	0	0
	AMP-TET-SXT	1.0	0	0.8	0	0	0
	AMP-TET-SOX	0.7	4.1	1.4	0	0	0
	AMP-SXT-SOX	0.7	0	0.5	0	0	0
	STR-TET-SOX	0	1.4	0.3	0	0	0
	STR-SXT-SOX	0	1.4	0.3	0	0	0
	TET-SXT-SOX	1.7	6.8	2.7	0	0	0
4	AMP-STR-TET-SOX	1.4	0	1.1	0	10.3	10.3
	AMP-STR-SXT-SOX	0.7	0	0.5	0	0	0
	AMP-TET-SXT-SOX	1.0	4.1	1.6	0	0	0
	STR-TET-SXT-SOX	0.3	1.4	0.5	0	0	0
5	AMP-STR-TET-SXT-ENR	0.3	0	0.3	0	0	0
	AMP-STR-TET-SXT-SOX	1.7	4.1	2.2	0	3.4	3.4
	AMP-STR-TET-ENR-SOX	0	0	0	0	3.4	3.4
	AMP-STR-SXT-ENR-SOX	0	1.4	0.3	0	0	0
	AMP-TET-SXT-ENR-SOX	0	2.7	0.5	0	0	0
6	AMP-STR-TET-SXT-ENR-SOX	0	0	0	25.0	0	6.9

**Table 4 antibiotics-13-00120-t004:** Antimicrobial susceptibility results of *E. coli* isolates from live bird markets (LBMs) (*n* = 169) and layer farms (LFs) (*n* = 43) using broth-based microdilution. Antimicrobials currently included in the Timor-Leste Antimicrobial Guidelines for humans are marked with an asterisk (*).

Antimicrobial Class/Antimicrobials	Percentage of Resistant Isolates (%)			MIC50 (mg/L)	MIC90 (mg/L)
	LBMs and LFs (*n* = 212)	From LBMs Only (*n* = 169)	From LFs Only (*n* = 43)		
**Penicillins**					
Ampicillin *	22.2	18.9	34.9	≤2	>8
Piperacillin	17.0	14.2	27.9	≤4	>64
Mecillinam	8.5	10.1	2.3	≤2	8
Temocillin	10.4	10.7	9.3	8	32
**β-lactam/β-lactam inhibitor combination**					
Amoxicillin-Clavulanic acid *	20.3	20.7	18.6	4	32
Piperacillin-Tazobactam *	0.5	0.6	0	≤4	≤4
**Cephalosporins**					
Cephalexin	4.7	3.0	11.6	8	16
Cefepime	5.2	5.9	2.3	≤1	≤1
Ceftriaxone *	5.2	5.9	2.3	≤0.5	≤0.5
Cefuroxime *	8.0	6.5	14.0	4	8
Ceftazidime	3.8	4.1	2.3	≤0.5	≤0.5
Cefixime	6.1	7.1	2.3	≤0.5	1
**Monobactams**					
Aztreonam	4.7	5.3	2.3	≤1	≤1
**Carbapenems**					
Ertapenem	0	0	0	≤0.25	≤0.25
Imipenem	0.9	0.6	2.3	0.5	1
Meropenem *	0	0	0	≤0.125	0.25
**Polymyxin**					
Colistin	6.6	6.5	7.0	≤0.5	1
**Aminoglycosides**					
Gentamicin *	5.7	5.9	4.7	2	2
Amikacin *	0	0	0	≤4	8
Tobramycin	4.7	4.7	4.7	2	2
**Fluroquinolones**					
Ciprofloxacin *	2.4	0.6	9.3	≤0.125	≤0.125
Levofloxacin *	1.4	0.6	4.7	≤0.5	≤0.5
**Sulfonamides**					
Trimethoprim-Sulfamethoxazole *	8.0	4.7	20.9	≤1	≤1
**Phosphonic acid**					
Fosfomycin w/G6P	0.9	0	4.7	≤16	≤16
**Nitrofuran**					
Nitrofurantoin *	0.5	0	2.3	≤16	32
**Multi-drug resistant**	13.7	11.8	20.9	-	-

**Table 5 antibiotics-13-00120-t005:** Antimicrobial susceptibility results of *Salmonella* spp. isolates (*n* = 17) from live bird markets (LBMs) and layer farms (LFs) using broth-based microdilution. Antimicrobials currently included in the Timor-Leste Antimicrobial Guidelines for humans are marked with an asterisk (*).

Antimicrobial Class/Antimicrobials	Percentage of Resistant Isolates (%)		
	LBMs and LFs (*n* = 17)	From LBMs Only (*n* = 1)	From LFs Only (*n* = 16)
**Penicillins**			
Ampicillin *	23.5	0	25.0
Piperacillin	23.5	0	25.0
Mecillinam	5.9	100	0
Temocillin	41.2	0	43.8
**β-lactam/β-lactam inhibitor combination**			
Amoxicillin-Clavulanic acid *	0	0	0
Piperacillin-Tazobactam *	0	0	0
**Cephalosporins**			
Cephalexin	0	0	0
Cefepime	0	0	0
Ceftriaxone *	0	0	0
Cefuroxime *	5.9	0	6.3
Ceftazidime	0	0	0
Cefixime	0	0	0
**Monobactams**			
Aztreonam	0	0	0
**Carbapenems**			
Ertapenem	0	0	0
Imipenem	5.9	100	0
Meropenem *	0	0	0
**Polymyxin**			
Colistin	5.9	100	0
**Aminoglycosides**			
Gentamicin *	5.9	100	0
Amikacin *	0	0	0
Tobramycin	5.9	100	0
**Fluroquinolones**			
Ciprofloxacin *	47.1	0	50.0
Levofloxacin *	17.7	0	18.8
**Sulfonamides**			
Trimethoprim-Sulfamethoxazole *	5.9	100	0
**Phosphonic acid**			
Fosfomycin w/G6P	0	0	0
**Nitrofuran**			
Nitrofurantoin *	0	0	0
**Multi-drug resistant**	5.9	100	0

**Table 6 antibiotics-13-00120-t006:** Crude associations between origin of *E. coli* isolates and antibiotic resistances from disk diffusion using mixed effect logistic regression models. Of the 367 *E. coli* isolates, 217 were from local chickens, 60 from fighting cocks, 16 from broilers and 74 from layer farms.

Antimicrobial/Origin of Isolate	Resistance (%)	Odds Ratio (95%CI)	*p* Value
Ampicillin			** *0.017* **
Local chicken	13.4	Ref	
Fighting cock	21.7	1.8 (0.9–3.7)	0.116
Broiler	43.8	5.2 (1.7–16.0)	**0.004**
Layer farm	23.0	2.0 (1.0–4.0)	0.066
Streptomycin			** *0.008* **
Local chicken	6.0	Ref	
Fighting cock	8.3	1.5 (0.5–4.4)	0.476
Broiler	31.3	8.2 (2.2–30.1)	**0.002**
Layer farm	16.2	3.3 (1.1–9.7)	**0.028**
Tetracycline			** *<0.001* **
Local chicken	12.9	Ref	
Fighting cock	18.3	1.5 (0.7–3.3)	0.287
Broiler	50.0	6.7 (2.3–19.4)	**<0.001**
Layer farm	48.6	6.4 (3.5–11.7)	**<0.001**
Enrofloxacin			** *0.003* **
Local chicken	0.9	Ref	
Fighting cock	0	N/A	N/A
Broiler	0	N/A	N/A
Layer farm	9.5	11.2 (2.3–55.4)	**0.003**
Trimethoprim/sulfamethoxazole			** *<0.001* **
Local chicken	6.5	Ref	
Fighting cock	8.3	1.3 (0.5–3.8)	0.611
Broiler	62.5	8.7 (2.8–27.4)	**<0.001**
Layer farm	29.7	4.7 (2.2–10.0)	**<0.001**
Sulfixosazole			** *<0.001* **
Local chicken	8.8	Ref	
Fighting cock	16.7	2.1 (0.9–4.8)	0.081
Broiler	62.5	17.4 (5.7–53.0)	**<0.001**
Layer farm	29.7	4.4 (2.2–8.8)	**<0.001**
Multi-drug resistant			** *<0.001* **
Local chicken	4.6	Ref	
Fighting cock	10.0	2.3 (0.8–6.5)	0.131
Broiler	43.8	18.1 (5.3–61.2)	**<0.001**
Layer farm	20.3	5.2 (2.0–13.1)	**0.001**

Overall likelihood ratio *p*-values are italicised, and individual Wald *p*-values are non-italicised. *p*-values < 0.05 are in bold. N/A—These categories are not included, as there were no isolates originating from fighting cocks or broilers that were resistant to enrofloxacin.

## Data Availability

The data presented in this study are available in the Appendix A.

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
