# Peer review of "Prevalence of Antimicrobial Resistance in Escherichia coli and Salmonella Species Isolates from Chickens in Live Bird Markets and Boot Swabs from Layer Farms in Timor-Leste"

_antibiotics, 2024, doi:10.3390/antibiotics13020120_

Round 1

Reviewer 1 Report

Comments and Suggestions for Authors

The research aimed to measure how widespread this resistance is in E. coli and Salmonella found in healthy chickens in Timor-Leste. Results indicated higher resistance to tetracycline and ampicillin. It also found that E. coli from commercial farms and broilers were more likely to be multi-drug resistant compared to local chickens, with specific concern over resistance to ciprofloxacin in Salmonella. Despite generally low resistance levels, the study recommends continuous monitoring due to potential growth in the commercial chicken industry and increased drug use. Here are few suggestions,

- While the introduction discusses the types of chickens and antibiotics used in Timor-Leste, it could benefit from clarifying if the different management practices of chicken types might influence AMR prevalence. It may be helpful to briefly compare AMR in Timor-Leste with regional data to highlight the country's unique challenges or similarities.

- Recommend further studies to explore the reasons behind the high prevalence of resistance to specific antimicrobials and the presence of MDR strains.

- The discussion on public health implications is prudent, especially the comparison with human AMR data. However, the potential for zoonotic transmission of AMR bacteria from chickens to humans could be explored further.

- Further discussion on how AMR in chickens compares to other sources of AMR within Timor-Leste would be beneficial. Additional studies should aim to gather direct data on antimicrobial usage in chicken populations to better understand resistance patterns.

Author Response

We thank the reviewer for taking time to provide comments and suggestions on this paper.  We have carefully revised the manuscript in line with these suggestions.

A “point by point” response to the reviewer’s comments is available in the document below.

Line numbers are provided when viewing the revised word document using “Simple Markup” and “All Markup”. Line numbers for “Simple Markup” version are in non-italicized, and line numbers for “All Markup” version are italicized)\.

Reviewer 2 Report

Comments and Suggestions for Authors

The authors here present a study of prevalence of antimicrobial resistance in E. coli and Salmonella isolates from chickens in LBMs and LFs in Timor Leste. The work presented here is both novel and relevant to the Journal as it is one of the first of its kind, conducted in a small developing Southeast Asian country. With chickens being the predominant players in the local animal farming and food production industry, the authors are interested in investigating the role of these poultry animals in transmission of AMR and potential public health and economic implications for Timor Leste. The authors do a good job explaining this and setting up a good background for their work in the introduction.

The methods of sample collection and further characterization are adequately described. The efforts put in by the authors into sample collection given the logistical, practical and technical issues is quite commendable.

The results section is well written and clearly organized, following a logical sequence of findings that corresponds to the methods section. The authors collected cloacal swabs from chickens in live bird markets and boot swabs from layer farms and tested them for susceptibility to six antimicrobials using disk diffusion and a subset for susceptibility to 27 antimicrobials using broth-based microdilution. Through their analysis, they found that E. coli and Salmonella spp. isolates were more resistant to tetracycline and penicillin classes of antimicrobials, which also were the most commonly imported and used in animals in Timor-Leste. THe authors also note that E. coli isolates from layer farms and broilers were more likely to be multi-drug resistant than those from local chickens. The authors also express their concern about resistance to colistin and ciprofloxacin detected in some isolates. 

The authors conclude that antimicrobial resistance in chickens is generally low in Timor-Leste, but there should be ongoing monitoring in commercial chickens as industry growth in their country might be accompanied with increased antimicrobial use. They also suggested developing antibiotic use guidelines for the poultry sector and strengthening the One Health approach to address antimicrobial resistance.

Some minor suggestions are made in the attached pdf on language and to clarify few interpretations. A few sentences should also be added to highlight the shortcomings of the study, especially the sample size and thus being cautious about the interpretation of data and comparisons between datasets. 

Comments on the Quality of English Language

The quality of English language in this manuscript is mostly good and just needs minor revisions here and there to meet the quality standards of MDPI journals. Most suggestions on English language have been made in the attached pdf and after necessary changes, the edited article should be ready for acceptance in my opinion.

Author Response

We thank the reviewer for numerous high-quality suggestions and comments. Most of the suggestions have been incorporated into the revised manuscript.

A “point by point” response to the reviewer’s comments is available in the attached document.

Line numbers are provided when viewing the revised word document using “Simple Markup” and “All Markup”. Line numbers for “Simple Markup” version are in non-italicized, and line numbers for “All Markup” version are italicized.

Reviewer 3 Report

Comments and Suggestions for Authors

Dear Editor

The manuscript "Prevalence of antimicrobial resistance in Escherichia coli and Salmonella species isolates from chickens in live bird markets and boot swabs from layer farms in Timor-Leste" described an emergent global issue. The Authors have presented this data in very good way. However, there are several shortfalls associated with the methodology of the current study. 

For example:

1. The methods lack molecular identification of the isolates i.e., sequence type analysis, WGS or PCR etc.

2. Antimicrobial resistance profiles are not supported by gene amplification or PCR etc.

3. 27 antibiotics are used in the current study. However, in reality there are several antimicrobial agents from same class i.e., Cephalosporins.

Thanks and Regards 

Comments on the Quality of English Language

The English Language is good and easy to understand.

Author Response

We thank the reviewer for taking time to provide comments and suggestions on this paper, and raising pertinent points that we have responded to. A “point by point” response to the reviewer’s comments is available in the attachment.

Round 2

Reviewer 3 Report

Comments and Suggestions for Authors

Dear Editor

The manuscript has been improved. Thanks and Regards

Comments on the Quality of English Language

The English language is well improved in the current format.